# Vaccinia Virus Arrests and Shifts the Cell Cycle

**DOI:** 10.3390/v14020431

**Published:** 2022-02-19

**Authors:** Caroline K. Martin, Jerzy Samolej, Annabel T. Olson, Cosetta Bertoli, Matthew S. Wiebe, Robertus A. M. de Bruin, Jason Mercer

**Affiliations:** 1MRC Laboratory for Molecular Cell Biology, University College London, London WC1E 6BT, UK; caroline.martin@einsteinmed.edu (C.K.M.); c.bertoli@ucl.ac.uk (C.B.); r.debruin@ucl.ac.uk (R.A.M.d.B.); 2Institute of Microbiology and Infection, University of Birmingham, Birmingham B15 2TT, UK; j.r.samolej@bham.ac.uk; 3School of Biological Sciences, University of Nebraska, Lincoln, NE 68583, USA; atolson@fredhutch.org; 4School of Veterinary and Biomedical Sciences, University of Nebraska, Lincoln, NE 68583, USA; mwiebe2@unl.edu

**Keywords:** vaccinia virus, cell cycle, kinase, p53, DNA damage response

## Abstract

Modulation of the host cell cycle is a common strategy used by viruses to create a pro-replicative environment. To facilitate viral genome replication, vaccinia virus (VACV) has been reported to alter cell cycle regulation and trigger the host cell DNA damage response. However, the cellular factors and viral effectors that mediate these changes remain unknown. Here, we set out to investigate the effect of VACV infection on cell proliferation and host cell cycle progression. Using a subset of VACV mutants, we characterise the stage of infection required for inhibition of cell proliferation and define the viral effectors required to dysregulate the host cell cycle. Consistent with previous studies, we show that VACV inhibits and subsequently shifts the host cell cycle. We demonstrate that these two phenomena are independent of one another, with viral early genes being responsible for cell cycle inhibition, and post-replicative viral gene(s) responsible for the cell cycle shift. Extending previous findings, we show that the viral kinase F10 is required to activate the DNA damage checkpoint and that the viral B1 kinase and/or B12 pseudokinase mediate degradation of checkpoint effectors p53 and p21 during infection. We conclude that VACV modulates host cell proliferation and host cell cycle progression through temporal expression of multiple VACV effector proteins. (209/200.)

## 1. Introduction

The cell cycle is the most fundamental molecular process of all life forms, orchestrating consecutive phases of cell growth (G1), DNA replication (S), DNA proofreading (G2), and cell division (mitosis, M) [1]. In mammalian cells, cell cycle progression, or movement through the various stages of the cell cycle, is driven by cyclin-dependent kinases (CDKs) and their specific activators, cyclins, while specialised molecular checkpoints ensure correct completion of each phase [1,2,3,4,5,6,7,8,9,10,11]. If a problem occurs during replication, such as DNA damage, checkpoint activation serves to pause the cell cycle and repair the damage or, if not possible, to induce apoptosis [2,4,12,13]. Together, the rate of cell division and cell death overtime define the net change in cell number, termed cell proliferation.

Given its central role in cell state, metabolic activity, availability of replication machinery and potential points of regulation, it is no surprise that viruses routinely target the host cell cycle to their own benefit [14]. By dysregulating cell cycle checkpoints—to arrest or induce cell cycle progression—many viruses modify the cell environment to promote viral genome replication, protein production and the assembly of progeny virions [14,15].

Vaccinia virus (VACV), the smallpox vaccine and prototypic poxvirus are amongst the viruses shown to regulate the host cell cycle upon infection. Like all members of the *Poxviridae*, VACV is a large, double-stranded DNA virus which replicates exclusively in the cytoplasm of host cells [16,17]. To achieve this, VACV encodes and expresses a large subset of viral DNA replication proteins including a DNA polymerase, a helicase–primase, a uracil DNA glycosylase, a processivity factor, a single-stranded DNA-binding protein, a DNA ligase and the replicative protein kinase, B1 [18,19].

Despite its exclusively cytoplasmic replication, VACV has been shown to inhibit cell cycle progression and mitosis [20,21,22,23]. At a molecular level, infection was shown to alter expression of CDKs, cyclins, and the tumour suppressors Rb and p53 in pre-synchronised cells during late infection [24,25,26]. To date, the only VACV protein connected to cell cycle regulation is the viral kinase B1 [27]. Overexpression of B1 in the absence of infection was found to mediate hyperphosphorylation of p53, thereby promoting its degradation. B1-mediated degradation of p53 was shown to require the E3 ligase Mdm2, which is the negative regulator of p53, and the proteasome [27,28,29]. While B1-mediated degradation of p53 has not been investigated in the context of VACV infection, VACV was found to upregulate *Mdm2* transcription, suggesting that a similar p53 degradation mechanism is employed during infection [26].

As a crucial effector in checkpoint signalling, p53 can pause the cell cycle and induce apoptosis. Cellular stress, such as DNA damage, promotes phosphorylation and stabilisation of p53 which then directs transcription of target genes, including the cell cycle inhibitor p21 that can arrest cells in G1/S and G2/M [2,8,30,31,32,33]. Conversely, inactivation of p53 prevents checkpoint-mediated cell cycle arrest in response to cellular stress, resulting in cell cycle dysregulation. Consistent with VACV-mediated dysregulation of the G1/S checkpoint through inhibition of p53 and Rb, a marked increase in S/G2 cells was observed during VACV infection [26].

Despite downregulation of these checkpoint effectors, VACV was shown to activate the cellular DNA damage response (DDR) to facilitate host protein-assisted viral genome replication [34]. The cellular DNA single-strand binding protein RPA was found to be recruited to replicating viral genomes, where it acts as a platform for replisome assembly. Within the replisome, the viral polymerase E9 was shown to interact with the DDR-activating protein topoisomerase II binding protein 1 (TOPBP1) and the cellular sliding clamp, proliferating cell nuclear antigen (PCNA). While activation of the DDR kinases ataxia telangiectasia and Rad3-related protein (ATR) and checkpoint kinase 1 (Chk1) was found to be essential, their role in VACV genome replication has not been established [34]. Additionally, mass spectrometry analysis of the proteins associated with replicating VACV genomes suggested that VACV might exploit several other cellular DNA repair pathways including Non-Homologous End Joining, Base Excision Repair, Nucleotide Excision Repair, Interstrand Crosslink Repair, and Homologous Recombination Repair (HRR) [35]. While neither ATR nor Chk1 was detected, PCNA was strongly enriched, as well as all subunits of the mini-chromosome maintenance MCM2-7 replicative helicase complex, the HRR components BLM, MRE11, NBS1, and Ku70, and topoisomerase I/II which was previously shown to facilitate VACV replication [36].

Collectively, these studies indicate that VACV infection triggers alterations in host cell cycle regulation and DDR to facilitate viral genome replication. However, the cellular factors and viral effectors that mediate these changes remain to be determined. By combining classical assays with state-of the art technologies, we set out to investigate the effect of VACV on cell proliferation and characterise the (viral) effectors required to dysregulate the host cell cycle. Confirming earlier studies, we found that VACV both inhibits and shifts the host cell cycle [20,21,22,25,26]. We further demonstrate that these two phenomena are independent of one another, with viral early genes responsible for inhibiting—and viral post-replicative gene(s) responsible for shifting—the cell cycle. Extending previous reports, we show that VACV B1 and/or B12 mediates degradation of checkpoint effectors p53 and p21 to facilitate the cell cycle shift followed by DNA damage checkpoint activation by the viral kinase F10 [25,26,27,34].

## 2. Materials and Methods

### 2.1. Cell Culture

HeLa (human, ATCC), HeLa fluorescent, ubiquitination-based cell cycle indicator (FUCCI, human, RIKEN Cell Bank, Ibaraki, Japan [37]), HeLa H2B-mCh (human, kind gift from M. Serres), HCT116 (human, kind gift from M. Wilson), and HCT 116 p53-/- (human, kind gift from M. Wilson) were grown in Dulbecco’s Modified Eagle Medium (DMEM, Life Technologies, Renfrew, UK) supplied with 10% FBS (Life Technologies, Renfrew, UK), 1% Gluta-Max (Gibco Fisher Scientific, Loughborough, UK), 1% Penicillin-Streptomycin (Life Technologies, Renfrew, UK), and 1% non-essential amino acids (Gibco Fisher Scientific, Loughborough, UK). BSC40 (African green monkey) medium additionally contained 1% sodium pyruvate (Life Technologies, Renfrew, UK). Cells were maintained under 37.0 °C, 5% CO_2_.

### 2.2. Antibodies and Reagents

The following antibodies were used at 1:1000 for Western blotting, unless otherwise indicated: anti-CDK1 (Cell Signaling #9116), anti-CDK2 (Cell Signaling #2546), anti-CDK4 (Cell Signaling #12790), anti-CDK6 (Cell Signaling #13331), anti-CDK7 (Cell Signaling #2916), anti-Chk1 (Santa Cruz sc-56291), anti-phospho-Chk1 Ser345 (Cell Signaling #2348), anti-Chk2 (Cell Signaling #3440), anti-phospho-Chk2 Thr68 (Cell Signaling #2197), anti-cyclinA (Santa Cruz sc-71682), anti-cyclinB1 (Cell Signaling #4138), anti-cyclinD1 (Cell Signaling #2978), anti-cyclinE (BD Pharmingen 551160), anti-F17 [38], anti-FLAG (Abcam ab205606), anti-I3 [39], anti-HA (BioLegend 902302, 1:2000), anti-HA (BioLegend 901502, 1:2000), anti-phospho-histone γH2A.X Ser139 (Cell Signaling #9718), anti-MCM2 (Cell Signaling #12079), anti-Mdm2 (Abcam ab16895), anti-mouse-HRP (Cell Signaling #7076, 1:5000), anti-mouse-IRDye 680RD (LI-COR 926-68072, 1:10,000), anti-mouse-IRDye 800CW (LI-COR 926-32212, 1:10,000), anti-p21 (Cell Signaling #2947), anti-p53 (Abcam ab1101), anti-rabbit-HRP (Cell Signaling #7074, 1:5000), anti-rabbit-IRDye 680RD (LI-COR 926-68073, 1:10,000), anti-rabbit-IRDye 800CW (LI-COR 926-32213, 1:10,000), anti-α-Tubulin (Cell Signaling #3873, 1:2000), and anti-α-Tubulin (Cell Signaling #2125, 1:2000).

Inhibitors were used at the following concentrations: Mevinolin (Lovastatin, 20 µM, LKT labs, St. Paul, MN, USA [40]), DL-Mevalonic acid lactone (Mevanolate, 6 mM, Sigma-Aldrich, Gillingham, UK), hydroxyurea (HU, 2.5 mM, Sigma-Aldrich, Gillingham, UK [41,42]), and RO3306 (10 µM, Sigma-Aldrich, Gillingham, UK). Mevinolin (Lovastatin) was converted to the active compound by dissolving the prodrug in 70% EtOH [43]. The function of each inhibitor is listed in Appendix A. Additionally, isopropyl β-D-1-thiogalactopyranoside (IPTG) was purchased from Sigma-Aldrich and used at the specified concentrations.

### 2.3. Viruses, VACV Purification, Titration and Infections

All viruses in this study were derived from the VACV Western Reserve (WR) strain. Apart from the wild type (WT WR), we used recombinant WR HA-D5 [44], WR F10-SH EL EGFP, and WR ΔB1mutB12 (a kind gift of M. Wiebe [45]); temperature-sensitive WR Cts24 (a kind gift of R. Condit [44,46]); IPTG-inducible vindH1 (a kind gift of P. Traktman [38]), and vL1Ri EL EGFP (a kind gift of B. Moss [47]). WT and recombinant viruses were produced in BSC40 cells, and purified as previously described [48]. The same protocol was followed for WR Cts24 but at permissive 31.0 °C. IPTG-inducible viruses were produced +/− IPTG (vindH1: 5 mM, vL1Ri EL EGFP: 50 µM). A description of each virus recombinant is listed in Appendix A.

Mature virion (MV) stocks were titred by plaque assay. Briefly, confluent BSC40 cells were infected with 10-fold serial dilutions of purified virus. After 48 h, cells were fixed and stained with 0.1% crystal violet in 3.7% PFA. Temperature-sensitive WR Cts24 was titred under permissive (31.0 °C) and non-permissive conditions (39.7 °C). IPTG-inducible viruses were titred in medium supplied with IPTG. As vindH1 and vL1Ri produced without IPTG (H1(−) and L1(−)) cannot be titred by plaque assay, virion concentration was determined indirectly through the DNA absorbance at 260/280 nm of the virus stock (NanoDrop, Thermo Fisher Scientific, Loughborough, UK ). The multiplicity of infection (MOI) equivalent (MOI eq.) was then calculated using the 260/280 nm absorbance of WT VACV with a known pfu/mL. Cells were infected with the specified viruses and MOIs in DMEM without FCS (Life Technologies, Renfrew, UK) for 1 h at 37.0 °C (Cts24: 31.0 °C, and 39.7 °C). The inoculum was then replaced with supplemented medium, and cells were incubated as indicated. For H1(+) and L1(+), both the infection and growth medium were supplied with the indicated concentration of IPTG.

### 2.4. Cell Proliferation Assay

Cells were counted using the automatic cell counter Cellometer (Nexcelom, Lawrence, MA, USA) and seeded into 6-well plates (Greiner Bio-one, Stonehouse, UK). After incubation overnight, samples were either mock infected or infected with the specified virus (MOI 5). Cells were harvested and cell counts were determined by two individual measurements per sample using the Cellometer (Nexcelom, Lawrence, MA, USA). To assess relative cell proliferation, cell counts were normalised to the respective baseline at 0 h post-infection (hpi).

### 2.5. Cytotoxicity Assay

Cytotoxicity was determined with the lactate dehydrogenase assay (Pierce^TM^ LDH Cytotoxicity Assay Kit, Thermo Scientific, Loughborough, UK) as per the manufacturer’s instructions. Cells grown to confluency in 96-well plates were either mock infected or infected with the indicated virus (MOI 5 and MOI 10). At the indicated time post-infection, the LDH assay was performed, and absorbance measured at 490 and 650 nm with a VersaMax Microplate Reader (Molecular Devices, Wokingham, UK) using SoftMax Pro software (Molecular Devices, Wokingham, UK). The 490 nm absorbance signal was background corrected by subtracting the absorbance at 650 nm. Values were normalised against the lysis control.

### 2.6. Western Blotting and Quantification

For immunoblots, cells were collected by scraping and centrifugation followed by resuspension in lysis buffer with protease inhibitors. Cell lysis was performed for 30 min on ice, samples were centrifuged at 20,000 g, 4 °C, for 10 min and the supernatant was boiled for 5 min in 4× loading dye with 4× DTT prior to loading on 4–12% Bis-Tris polyacrylamide gels (Thermo Fisher Scientific, Loughborough, UK ). After transfer to nitrocellulose, membranes were blocked with 5% milk in TBS-T (Sigma-Aldrich, Gillingham, UK) for 1 h before blotting. Primary antibodies were diluted in blocking solution and incubated overnight at 4 °C. Membranes were washed and incubated with HRP, or IRDye-secondary antibodies in blocking solution for 2 h at RT, washed and analysed with ImageQuant LAS 4000 Mini (GE Life Sciences through Sigma-Aldrich, Gillingham, UK) and Luminata Forte Western HRP Substrate (Merck, Darmstadt, Germany) for detection. IRDye secondary antibodies were imaged with a LiCOR ImageQuant (GE Life Sciences, through Sigma-Aldrich, Gillingham, UK). Independent of the detection method, protein intensities were quantified using the software StudioLite (LiCOR, Cambridge, UK). Where applicable, samples were normalised against an internal α-Tubulin loading control.

### 2.7. Cell Cycle Analysis by Flow Cytometry

Asynchronous, subconfluent HeLa FUCCI cells were either mock infected, or infected with the specified virus (MOI 2). Cells were washed, harvested and analysed using a GUAVA easyCyte HT (Merck Millipore, Darmstadt, Germany). Data were processed with InCyte 3.1.1 (Merck Millipore, Darmstadt, Germany), scoring cells as either early G1, late G1, early S, or S/G2/M (Appendix A [37]).

For synchronisation/release experiments, HeLa FUCCI cells were either arrested in G1 with Lovastatin (20 µM), or in the S phase with HU (2.5 mM), or in G2 with RO3006 (10 μM), as previously described [43,49]. Immediately before infection, cells were released from the block by washing twice with PBS. For Lovastatin, release medium was supplemented with Mevanolate (6 mM). The samples were then prepared for flow cytometry and analysed as above.

### 2.8. Visualisation of Cellular DNA Synthesis Assay

HeLa Kyoto H2B-mCh cells seeded on coverslips were either mock infected or infected with WR HA-D5 (MOI 8). At 15 min before fixation, cells were fed with 200 μL 4× EdC (40 μM, final = 10 μM). Cells were washed with PBS and fixed with 4% PFA at RT for 15 min. Fixed samples were washed 3× with PBS and stained for EdC using the Click-iT^TM^ EdU Alexa FluorTM 488 Imaging Kit (Invitrogen), following the manufacturer’s instructions. Coverslips were mounted with Immu-Mount (Thermo Scientific, Loughborough, UK), and a minimum of 5 different locations per coverslip were imaged with a 100× oil immersion objective (NA 1.45) on a VT-iSIM microscope (Visitech; Nikon Eclipse TI, Sunderland, UK). Nuclei were counted manually using ImageJ.

### 2.9. Overexpression of Viral Proteins

Cells were transfected with 2 µg of indicated plasmid DNA using Lipofectamine 2000 (Invitrogen, Thermo Scientific, Loughborough, UK) following the manufacturer’s instructions.

### 2.10. SiRNA Knockdown of Viral Proteins

Cells were reverse transfected with the indicated siRNAs using Lipofectamine RNAiMAX (Invitrogen, Thermo Scientific, Loughborough, UK) following the manufacturer’s instructions. Briefly, siRNA (stock 10 µM) and RNAiMAX were individually diluted in DMEM(-), combined and incubated for 1 h at RT. In total, 250,000 cells in suspension were added to the siRNA/RNAiMAX solution (final siRNA concentration of 20 nM) and plated. Transfected cells were grown for 48 h before infection. The following siRNA sequences with a dTdT modification (Sigma-Aldrich, Gillingham, UK) were used: siA24R 5′-CUGCUAAGCCGUACAACAA-3′ [44], siF10L 5′-AACUGGUAUUACGAUUUCCAUU-3′, Scrambled (allStars Negative, Qiagen, Manchester, UK).

## 3. Results

### 3.1. VACV Infection Inhibits Cell Proliferation

Cell proliferation is defined as the net change in cell number over time due to cell division and cell death. VACV infection has been reported to inhibit cell proliferation of mouse fibroblasts, and to have a pro-proliferative effect in human 143B osteosarcoma cells [26]. As this suggested that the effects of VACV infection on proliferation may be cell-type specific, we asked how VACV infection impacted proliferation of HeLa, BSC40 and HCT116 cells. Mock-infected cells and cells infected with VACV were assessed at 0, 24 and 48 h post-infection (hpi) for cell number and cell death. While mock-infected cell numbers doubled every 24 h as expected, VACV-infected cell numbers did not increase over the 48 h time course in any of the cell lines (Figure 1A). The accompanying cytotoxicity assays showed that mock- and VACV-infected cells displayed the same level of cytotoxicity over 48 h, even when doubling the MOI (Figure 1B). These results indicated that VACV infection reduces cell proliferation through inhibition of cell division as opposed to induction of cell death.

### 3.2. VACV-Mediated Inhibition of Cell Proliferation Requires Virus Fusion but Not Genome Uncoating

As illustrated in Figure 1C, VACV enters host cells by macropinocytosis [48,50]. Once internalised, virions fuse from the macropinosome and deposit viral cores and their associated lateral bodies (LBs) into the host cytoplasm [51]. Pre-replicative early gene expression is initiated, allowing for the production of proteins required for subsequent genome uncoating [44], DNA replication and post-replicative intermediate and late gene expression [19]. Using recombinant VACVs defective for virus entry or genome uncoating, we sought to define the stage of the virus life cycle required for inhibition of cell division. To test whether VACV entry was required, we used a recombinant VACV (vL1i), which is inducible for the expression of the viral protein L1 [47]. The L1 protein is a member of the VACV entry–fusion complex, thus virions produced in the absence of L1, L1(−) virions are unable to mediate membrane fusion [47,52]. When BSC40 cells were infected with L1(+) virions, cell proliferation was inhibited as expected (Figure 1D). However, when cells were infected with L1(−) virions, they continued to multiply at a level similar to mock-infected cells, accumulating 2.7-fold over 48 h. This result indicated that a post-fusion cytoplasmic stage of the VACV lifecycle is required to inhibit cell proliferation.

We next asked if inhibition of host cell proliferation occurred pre- or post-VACV genome uncoating using a VACV strain which is temperature sensitive (ts) for the expression of the viral uncoating factor D5 (Cts24) [18,44,53]. HeLa cells were infected with Cts24 under permissive (31 °C) and non-permissive (40 °C) uncoating conditions and cell proliferation was assessed over 48 h. We found that VACV-mediated inhibition of cell proliferation was unabated under non-permissive uncoating conditions (Figure 1E). This indicated that neither D5, nor the process of viral genome uncoating is required to block cell proliferation. Taken together, an event between VACV core deposition and genome uncoating is required to inhibit the host cell cycle.

### 3.3. VACV Infection Induces a General Cell Cycle Arrest

Having found that VACV inhibits cell proliferation, we next asked at which cell cycle stage cells arrest upon infection. The cell cycle is divided into four consecutive phases: cell growth (G1), DNA replication (S), DNA repair (G2) and division (M). Cells that exit the cell cycle enter quiescence (G0), a dormant state marked by the absence of cell proliferation, CDK activity and reduced levels of the DNA replication licensing factor MCM2 [54,55,56,57,58,59].

Given the potent inhibition of cell proliferation observed upon VACV infection, we first asked if VACV was driving cells into quiescence. We compared MCM2 protein levels between mock-infected control and VACV-infected cells over a 24 h time course (Figure 2A). We found that infection did not significantly alter MCM2 levels compared to mock-infected cells (Figure 2B). This suggested that VACV does not drive cells into quiescence but rather inhibits an active stage of the cell cycle.

To identify which stage(s) were inhibited, we used HeLa FUCCI cells, which allowed us to distinguish four different cell cycle phases based on fluorescence: early G1 (no fluorescence), G1 (red only), early S (red and green), and S/G2/M (green only) (Appendix A) [37]. These cells were synchronised in either G1 with Lovastatin [60,61], S phase with hydroxyurea [62,63], or in late G2 with the CDK1 inhibitor RO3306 [64] (Appendix A). Cells were released, infected with VACV and the cell cycle distribution—relative to mock-infected cells—determined at 24 hpi. As expected, under all synchronisation and release conditions (Lovastatin, hydroxyurea and RO3306, all from Sigma-Aldrich, Gillingham, UK), mock-infected cells re-entered the cell cycle, resulting in reduced G1, S, and G2 fractions, respectively (Figure 2C–E; black bars). Conversely, a portion of cells infected with VACV were retained in the pre-synchronised stage independent of the synchronising agent: relative to mock-infected cells, 12% more VACV-infected cells remained in G1 after release from Lovastatin (Figure 2C), 28% more remained in S/G2/M after release from the hydroxyurea S-phase block (Figure 2D) and 22% more remained in S/G2/M after release from the G2 block mediated by RO3306 (Figure 2E). These results show that VACV inhibits G1, S, G2 and/or M-phase progression, suggesting that infection establishes a general cell cycle block.

As we found that a stage of the VACV lifecycle between fusion and uncoating inhibits cell proliferation (Figure 1), and viral early gene expression (EGE) occurs between these two, we asked if EGE was required for the general block in cell cycle we observed. To address this, we used a recombinant VACV, vindH1, which is inducible for the expression of the viral H1 phosphatase [38]. Virions produced in the absence of inducer (H1(−)) are attenuated for early gene expression [38,65]. As above, HeLa FUCCI cells were synchronised in G1, S or late G2, released and infected with H1(+) or H1(−) virions. Cells were then assessed for their cell cycle distribution relative to WT-infected cells. Cells infected with H1(+) virions, which are akin to WT virions, largely remained in their pre-synchronised cell cycle stage, while those infected with H1(−) virions were no longer capable of blocking cell cycle re-entry (Figure 2F–H). Relative to H1(+) virions, 11%, 45%, and 29% of cells re-entered the cell cycle after release from the G1, S or late G2 blocks, respectively. This indicates that the H1 phosphatase, a known immune modulator [51,66]—or viral EGE—is required for blocking cell cycle progression upon VACV infection.

### 3.4. VACV Infection Inhibits Cellular DNA Synthesis

While we found that VACV can retain cells in G1 and S/G2/M, the FUCCI system did not allow for further resolution of the individual effect of VACV on S, G2 and M. Therefore, we next addressed whether VACV could inhibit M- and/or S-phase progression. To assess M phase during VACV infection, cells were infected with WT VACV, fixed, and stained with the DNA dye Hoechst at various time points between 0 and 24 hpi. Using genome condensation as a visual marker, we quantified the number of mitotic infected cells relative to the number of mitotic mock-infected cells at each time point (Figure 3A). While the percentage of mitotic cells fluctuated between 1.3% and 8.1%, the number of mitotic cells in infected samples declined until no mitotic cells were observed by 24 hpi (Figure 3A). This indicated that VACV is capable of blocking M phase either by preventing entry into mitosis, or by arresting cells prior to DNA condensation.

We next addressed the impact of VACV infection on S phase, during which the cellular genome is replicated. It has been reported that VACV inhibits cellular DNA replication in HeLa cells [20]. To confirm and extend these findings, we established a microscopy-based cellular DNA replication assay using the nucleotide analogue EdC, which we have shown readily incorporates into cellular DNA, but not VACV replication sites [67]. By pulse-labelling mock and VACV-infected HeLa cells that express histone H2B-mCherry for 15 min prior to fixation and labelling, we could quantify the percentage of cells undergoing active cellular DNA synthesis (illustrated in Appendix A). When the percentage of EdC-positive infected cells was quantified over time we found that VACV infection completely blocked cellular DNA synthesis by 24 hpi, consistent with previous reports (Figure 3B). Inhibition of cellular DNA synthesis appeared to be due to an early event in the virus lifecycle with a 44% and 84% reduction in EdC-positive infected cells, relative to controls at 4 and 8 hpi, respectively.

### 3.5. Inhibition of Cellular DNA Synthesis Is Independent of Viral Genome Replication

To determine which stage of the virus lifecycle was required for the observed inhibition of cellular DNA synthesis, we repeated this assay with viruses defective for either virus fusion (vL1i) or genome uncoating (Cts24) as described in Figure 1. We found that fusion competent L1(+) virions abolished EdC incorporation by 24 hpi, while fusion-incompetent L1(−) virions appeared to have no impact on cellular DNA synthesis relative to controls (Figure 3C). As observed for inhibition of cell proliferation (Figure 1), a post-fusion cytoplasmic stage of the virus lifecycle is also responsible for inhibition of cellular DNA synthesis.

Using Cts24, we next asked if genome uncoating was required to inhibit cellular DNA synthesis. Infection under both permissive uncoating (Cts24 31 °C) and non-permissive uncoating (Cts24 40 °C) conditions resulted in complete inhibition of EdC incorporation by 24 hpi (Figure 3D). This indicated that neither viral genome uncoating, nor viral genome replication was required to block cellular DNA synthesis. Collectively, these results suggest that viral core deposition and/or a lateral body constituent are required to block cellular replication.

### 3.6. VACV Post-Replicative Gene Expression Shifts the Host Cell Cycle

Our results indicate that VACV blocks host cell proliferation, arresting cells in G1, S, and G2, and inhibits both mitosis and DNA replication (Figure 1, Figure 2 and Figure 3). This suggested to us that VACV either inhibits the cell cycle systemically (i.e., upon infection cells remain in the cell cycle stage they are in), or that VACV arrests DNA replication and mitosis, thereby effectively trapping cells in either G1 or G2 phases of the cell cycle. To test this, we infected unsynchronised HeLa FUCCI cells with WT VACV and followed the cell cycle distribution over 24 h (Figure 4A). Relative to mock-infected cells, those infected with VACV gradually shifted from G1 into S/G2/M, resulting in a significant increase in the S/G2/M fraction at the expense of G1 by 24 hpi. Having established that infection inhibits cellular DNA replication (S phase) and mitosis (M phase), this result suggests that infected cells are shifted out of G1 and likely accumulate in S/G2.

To define which step of the virus life cycle was required to shift the host cell cycle out of G1, we tested the requirement for viral EGE by following the cell cycle distribution of unsynchronised HeLa FUCCI cells infected with H1(+) or H1(−) VACV (Figure 4B). As expected, H1(+)-infected cells were gradually shifted out of G1, resulting in accumulation in S/G2 by 24 hpi. In the absence of H1, VACV could no longer shift cells from G1 to S/G2, indicating that viral EGE and/or H1 was required to shift the host cell cycle. To narrow in, we asked if post-replicative viral gene expression could mediate the cell cycle shift. For this, we depleted a component of the VACV DNA-dependent RNA polymerase, A24, using virus targeting siRNA [44]. As the VACV DNA-dependent RNA polymerase is pre-packaged in virions to direct in-core viral early gene transcription, siRNA depletion of A24 only affects post-replicative intermediate and late gene transcription which rely on newly expressed A24. The effectiveness of this approach was demonstrated by monitoring the expression of an early (I3) and late gene (F17) after A24 depletion (Figure 4C). Relative to mock-infected cells treated with control siRNA, WT VACV infection had shifted the distribution of cells from G1 into S/G2 by 24 h as expected (Figure 4D; Scr WT). Depletion of A24 prevented the shift from G1 to S/G2 in WT-infected samples (Figure 4D; siA24R WT). Compared to scrambled WT VACV controls, the G1 fraction was significantly increased, and the S/G2 fraction significantly decreased when viral post-replicative gene expression was inhibited. These findings suggest that viral post-replicative gene expression is required to shift the host cell cycle from G1 to S/G2 late in infection.

### 3.7. The Viral Kinase F10 Activates the Cellular DDR Late in Infection

To gain additional insight into how VACV arrests and shifts the host cell cycle, we turned to the underlying cell signalling pathways that control these processes. Cyclin-dependent kinases (CDKs) and their activators, the cyclins, are the key drivers of cell cycle progression. Periodic expression of cyclins targets the activity of their partner CDK to defined phases of the cell cycle. It has been shown that VACV alters CDK and cyclin expression in synchronised rabbit fibroblasts, as well as the transcript levels of cyclins in unsynchronised HeLa cells [24,26]. To test if VACV altered the expression of CDKs and cyclins at the protein level in unsynchronised HeLa cells, we analysed CDK 1, 2, 4, 6, 7 and cyclins A, B, D, and E by immunoblot analysis over 24 h (Appendix A). Contrary to previous findings with synchronised cells [25], neither CDK (Appendix A) nor cyclin (Appendix A) protein levels were found to be differentially regulated during the first 24 h of infection (Appendix A).

As these results indicate that VACV does not prevent proliferation by depleting positive regulators, we next focused on cell cycle inhibitors. Molecular checkpoints can either pause the cell cycle or induce apoptosis upon aberrant cell cycle events such as DNA damage. The kinases ATR and/or ATM detect DNA breaks and trigger the cellular DNA damage response (DDR) through phosphorylation of the kinases Chk1 and/or Chk2 [12,13,68]. As VACV has been reported to activate the DDR in a pre-uncoating step [34], we asked if DDR was the trigger for cell cycle arrest during early infection. First, we monitored DDR activation during VACV infection using phosphorylation of Chk1 (Ser345) and Chk2 (Thr68) as a readout. Immunoblot analysis of WT VACV-infected HeLa cells showed Chk1 Ser345 phosphorylation after 8 hpi, and Chk2 Thr68 phosphorylation from 4 hpi, with a strong signal increase after 6 hpi (Figure 5A,B). DDR activation did not occur in mock controls (Appendix A), and was not due to cellular DNA damage as determined by immunoblots directed against phosphorylation of γH2AX Ser139 (Figure 5A). While these findings confirmed robust DDR activation in response to VACV infection, the kinetics of the response under our experimental setting was not consistent with a pre-uncoating event as the trigger [34].

The late phosphorylation of Chk1 and Chk2 suggested that a post-replicative event in the VACV life cycle activated the DDR. We therefore tested the requirement for viral intermediate and late gene expression by silencing A24 (siA24R). HeLa cells were infected with a recombinant VACV strain, WR F10-SH EL EGFP, that expresses a C-terminal streptavidin-HA tagged version of F10 from its endogenous locus and EGFP under a viral Early/Late promoter from the TK locus. While the scrambled control siRNA did not affect VACV-mediated Chk2 activation, siA24R strongly reduced Chk2 Thr68 phosphorylation in infected cells (Figure 5C). This implicated a viral intermediate/late protein in Chk2 activation. As activation is driven by phosphorylation, we hypothesised that the late VACV-encoded kinase, F10 [69,70], may be responsible. F10 was depleted using siRNA (siF10L) and the knockdown confirmed by immunoblot directed against the SH tag (Figure 5C; F10-SH). We found that depletion of F10 prevented Chk2 Thr68 phosphorylation, suggesting that F10 is required for DDR activation.

To determine if F10 was sufficient to trigger Chk2 phosphorylation, or if it requires other viral factors, we expressed codon-optimised version of 3 × FLAG tagged F10 (3 × FLAGco) in uninfected cells and monitored Chk2 Thr68 phosphorylation. Expression of 3 × FLAG-F10 resulted in a 2.4-fold increase in Chk2 Thr68 phosphorylation over the GFP vector control (Figure 5D). These results show that VACV F10 is sufficient to trigger phosphorylation and activation of the DDR kinase Chk2 in the absence of infection.

### 3.8. VACV Induces Degradation of p53 and p21

Both DDR and the G1/S checkpoint pause the cell cycle through activation of effectors such as p53 and its transcriptional target, the CDK inhibitor p21 [30,33,71]. Under homeostatic conditions, the p53-specific E3 ligase, Mdm2, ubiquitinates p53, resulting in its degradation [28,29]. In stressed cells, activation of p53 and accumulation of p21 arrest the cell cycle through CDK inhibition, and further prevent cellular DNA replication through inhibition of the DNA-polymerase processivity factor PCNA [72]. Not surprisingly, modulation of p53/p21 activity and expression is a well-documented viral strategy to create a pro-viral environment [14,15]. VACV has been shown to reduce p53 levels during late infection in pre-synchronised cells [25,26].

We therefore asked if VACV infection altered Mdm2, p53 and p21 levels independent of DDR activation using immunoblot analysis of Mdm2, p53 and p21 in mock and WT-infected cells over 24 h (Figure 6A,B). In mock-infected cells, Mdm2, p53, and p21 protein levels fluctuated but were detectable during our experiment (Appendix A). Compared to mock-infected cells, VACV caused a rapid reduction in Mdm2 proteins levels by 4 hpi followed by downregulation of p53 and p21 by 6 hpi (Figure 6B and Appendix A). Of note, the downregulation of Mdm2, p53 and p21 all occurred prior to late viral gene expression (Figure 6B; F17).

This finding implied that VACV does not arrest the cell cycle via upregulation of p53 and/or p21 expression. To exclude that p53 arrests the cell cycle prior to its downregulation at 6 hpi, we asked if VACV could inhibit proliferation of HCT116 cells deleted of p53 (HCT116 p53-/-). HCT116 control and HCT116 p53-/- cells were mock infected or infected with WT VACV and cell numbers quantified at 24 and 48 hpi (Figure 6C). As with BSC40 and HeLa cells, we observed a complete inhibition of cell proliferation upon infection of either cell line. These results confirmed that p53 is dispensable for VACV-induced cell cycle arrest.

### 3.9. VACV ΔB1mutB12 Fails to Decrease Cellular p53, and p21 Levels

We next looked to identify the viral protein(s) that facilitate degradation of Mdm2, p53, and p21. In uninfected cells, phosphorylation of p53 at Ser15, Thr18, and Ser20 controls its interaction with its negative regulator Mdm2 [73,74]. As dynamic (de)phosphorylation regulates p53 stability, we hypothesised that a viral kinase or phosphatase may destabilise p53. The viral early kinase B1 was a promising candidate as p53 was degraded prior to late gene expression. In addition, B1 shares homology with the cellular kinase VRK1 (vaccinia-related kinase 1), which modulates p53 stability through Thr18 phosphorylation [75], and expression of B1 in uninfected cells was shown to be sufficient to phosphorylate p53 at its N-terminus, which was suggested to promote p53 binding to Mdm2 and proteasome-dependent degradation [27].

As B1 is essential for viral genome replication, siRNA silencing or genetic deletion (ΔB1) abrogates viral DNA synthesis and the production of VACV particles [46,76,77]. However, genetic truncation of the viral pseudokinase B12 in the ΔB1 virus (ΔB1mutB12) rescues viral replication through an incompletely understood mechanism involving the cellular VRK1 protein [45]. Therefore, VACV ΔB1mutB12 allowed us to ask if B1 and/or B12 directed the downregulation of Mdm2, p53, and p21, independent of B1′s function in viral replication. Immunoblot analysis showed that, like WT VACV, ΔB1mutB12 caused a rapid decease in Mdm2 levels; however, it failed to reduce the levels of p53, and p21 (Figure 6D and Appendix A). This indicates that B1 and/or B12 mediate the degradation of p53, and p21 during VACV infection, whereas the reduction of Mdm2 is independent of these viral factors.

Next, we asked if B1/B12 were required to block the host cell cycle, despite their role in degrading p53 and p21. As before, HeLa cells were infected with WT or VACV ΔB1mutB12 and the cell number, relative to mock-infected cells, quantified at 24 and 48 hpi. ΔB1mutB12 blocked cell proliferation to the same level as WT VACV, confirming that B1, B12 and downregulation of p53 and p21 are—as expected—not necessary for the VACV-induced cell cycle arrest (Figure 6E).

## 4. Discussion

In this study, we investigated the impact of VACV infection on the host cell cycle in cancer cells. We show that VACV causes a general cell cycle arrest and shifts cells from G1 into S/G2. Our data suggest that these effects are two independent functions of viral early, and post-replicative gene expression, respectively. Looking for the underlying molecular mechanism, we found that the viral kinases B1 and F10 modulate the cell cycle checkpoint machinery. F10 was found to trigger the DNA damage checkpoint and B1/B12 caused degradation of the checkpoint effectors p53 and p21.

To characterise how VACV arrests the cell cycle, we used WT VACV and a set of viral mutants that block at defined stages of the virus lifecycle. Consistent with previous reports, we showed that VACV infection triggers a general cell cycle block between virus fusion and genome uncoating [20]. Using a virus that lacks H1 phosphatase, we pin-pointed this block to either H1 within virions or viral early gene expression. It has been reported that UV-irradiated VACV cannot trigger cell cycle arrest [72]. Given that UV-irradiated virus only expresses a subset of early genes, and pre-packaged H1 should not be impacted by UV treatment, it is likely that an early gene(s) and not H1 mediates the cell cycle arrest.

We reasoned that the viral effector(s) could prevent cell cycle progression by inhibiting a positive cell cycle regulator, such as CDKs or cyclins, or conversely by activating a negative cell cycle regulator, such as a cell cycle checkpoint. We found that VACV infection had no impact on the protein levels of cyclins A, B, D, and E and CDKs 1, 2, 4, 6, and 7 up to 24 hpi, despite transcriptomic analysis showing decreased mRNA levels for cyclins A2, C, D1, G1, and H1 at 16 hpi in HeLa cells [24]. In general agreement with our data, proteomic analysis indicated that cyclin A2, B1, B2, H, and CDKs 1, 2, 6, 7 levels were not altered by infection [73]. Taken together, these data strongly suggest that VACV infection does not inhibit cell proliferation by actively diminishing cyclin and/or CDK protein levels. These data also suggest that VACV-mediated cell cycle arrest is specifically driven by an early viral gene, and not due to a general effect, such as virus-mediated host shut-off [74,75,76].

Next, we addressed if VACV cell cycle inhibition was triggered by activation of a cell cycle checkpoint. Complementing the function of cyclins and CDKs, these checkpoints can pause the cell cycle in case of stress signals such as cellular DNA damage. As VACV infection was reported to activate the cellular DNA damage response (DDR) before viral genome uncoating [34], we tested if this caused the cell cycle arrest. Using Chk1 and Chk2 phosphorylation as a readout of DDR activation, we found that DDR was triggered during the late stages of infection (>10 h), and that Chk1 and Chk2 phosphorylation were dependent upon the post-replicative viral kinase F10. It remains to be determined how F10 triggers DDR and how this relates to early VACV cell cycle arrest.

To test if VACV infection triggered a DDR-independent checkpoint, we next focused on the effector p53 and its transcriptional target, the cell cycle inhibitor p21 [2,8,30,31,32,33]. In line with previous studies, we found that VACV infection resulted in depletion of p53 and p21 in a B1/B12-dependent fashion [25,27,70]. Contrary to previous reports [26,27], we found that Mdm2, a negative regulator of p53 stability, was also degraded. This suggests that VACV uses an alternative, Mdm2-independent strategy for p53/p21 degradation during infection. Nonetheless, using p53-/- HCT116 cells and a B1 deletion virus, we conclusively show that neither B1, nor p53 is required for the VACV-induced cell cycle arrest. As checkpoint activation can also induce p53-independent arrest [32,52,77], these findings do not exclude that VACV inhibits the cell cycle through triggering a checkpoint.

While p53 degradation does not explain the cell cycle arrest, it might facilitate the VACV-induced cell cycle shift. Consistent with previous observations in cells released from a G1 block, we demonstrate that VACV post-replicative gene expression shifted cells into S/G2 at the expense of the G1 and M phases [25], [26]. For cells to be able to transition from G1 into S/G2, they need to be released from the initial cell cycle block, established during early infection. Our data are in line with a model in which VACV allows S-phase entry by disassembling the G1/S checkpoint through deactivation of p53, p21 and Rb [26]. That p53 and p21 are degraded before post-replicative gene expression suggests that deactivation of the G1/S checkpoint is not sufficient to shift infected cells. In agreement with our observation that cell cycle activators (CDKs and cyclins) are not reduced, our model presumes that infected, arrested cells still have the ability to progress through the cell cycle. Together, these findings indicate that viral early gene expression triggers a general cell cycle arrest, then specifically relieves the G1/S checkpoint before post-replicative gene expression actively promotes a G1 to S/G2 transition (Figure 7).

Taken together, we show that VACV inhibits the host cell cycle prior to genome uncoating, independent of the tumour-suppressor p53, its transcriptional target p21 and DDR activation. We suggest that early degradation of p53 and p21 serves to deactivate the G1/S checkpoint, allowing an unidentified viral late gene to shift the host cell cycle from G1 into S/G2. Future work will be aimed at identifying the viral “shift” protein and determining if the initial cell cycle arrest is a direct effect of infection, or a more general effect of virus-induced host shut-off. Furthermore, the function of the observed cell cycle changes in the virus life cycle will be addressed.

## Figures and Tables

**Figure 1 viruses-14-00431-f001:**
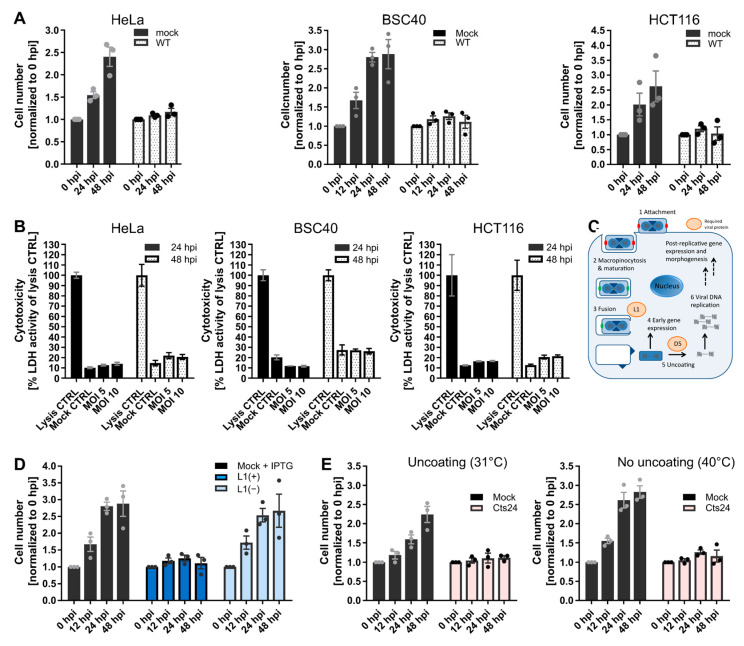
VACV inhibits the host cell cycle. (**A**) HeLa, BSC40, and HCT116 cells were either mock infected, or infected with WT VACV at MOI 5 and cell numbers were determined over 48 h. (**B**) HeLa, BSC40, and HCT116 cells were either mock infected, or infected with WT VACV at MOI 5, and MOI 10. Cytotoxicity was measured using the Pierce LDH Cytotoxicity Assay Kit at 24 and 48 h and normalised to the cell lysis control. Data represent two biological replicates of technical triplicates and are displayed as the mean ± S.D. (**C**) VACV life cycle schematic highlighting the viral proteins L1 and D5 which are required for fusion and uncoating, respectively. The inactive viral fusion machinery is represented by red dots; its fusion-competent form by green dots. Black arrows indicate the respective next step in the virus life cycle. (**D**) BSC40 cells were either mock infected, or infected with VACV L1(+), or VACV L1(−) at MOI 5 and cell numbers were determined over 48 h. (**E**) BSC40 cells were either mock infected, or infected with VACV Cts24 at MOI 5 and incubated at either the permissive (31 °C), or non-permissive temperature (40 °C) and cell numbers were determined during 48 h. (**A**,**D**,**E**) Data represent biological triplicates, each with technical duplicates, and are displayed as the normalised mean ± S.E.M. Grey and black dots represent means of individual biological replicates.

**Figure 2 viruses-14-00431-f002:**
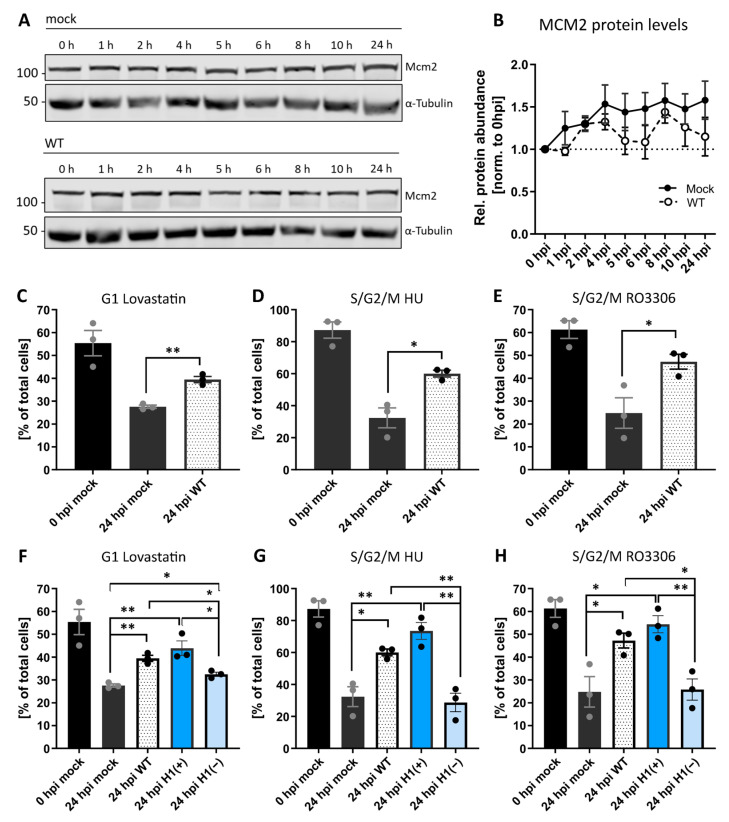
VACV establishes a general cycle block. (**A**) Immunoblot analysis of MCM2 during WT VACV infection. HeLa cells were either mock infected or infected with WT VACV at MOI 5 and harvested over 24 h. Whole-cell lysates were resolved via SDS-PAGE, and immunoblotted for MCM2, and α-Tubulin as loading control. A representative blot of three replicates is shown. (**B**) MCM2 protein abundance was quantified and normalised to the loading control. Data represent biological triplicates and are displayed as the mean ± S.E.M. [C–H] HeLa FUCCI cells were synchronised in the indicated cell cycle stage with either Lovastatin, HU, or the CDK1 inhibitor RO3306. Immediately after release, cells were mock infected (black), or infected with WT (white), H1(+) (blue), or H1(−) (light blue) VACV at MOI 2. The cell cycle distribution was assessed at 0, and 24 hpi by flow cytometry. (**C**,**F**) Percentage of G1 cells. [D–E, G–H] Percentage of S/G2/M cells. All data represent biological triplicates and are displayed as the mean ± S.E.M. Grey and black dots represent individual biological replicates. Data in (**C**–**E**) are replicated in (**F**–**H**). Parametric, unpaired, two-tailed t-test for significance. ns. *p* > 0.05, * *p* < 0.033, and ** *p* < 0.0021.

**Figure 3 viruses-14-00431-f003:**
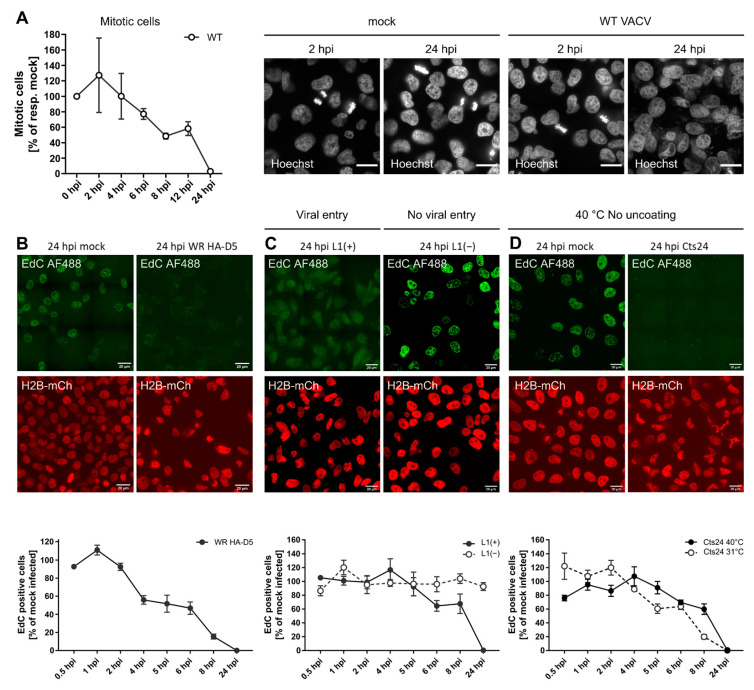
VACV inhibits mitosis and cellular DNA synthesis. (**A**) To quantify mitotic cells during VACV infection, HeLa cells were either mock infected, or infected with WT VACV at MOI 5, fixed and stained with the DNA dye Hoechst at various time points between 0 and 24 hpi. Using genome condensation as a visual marker, the number of mitotic infected cells was quantified relative to the number of mitotic mock-infected cells at each time point. Representative images are shown. Scale bar = 20 μm. Plotted data represent biological triplicates and are displayed as the mean ± S.E.M. (**B**–**D**) Pulse labelling with the nucleotide analogue EdC was used to identify active cellular DNA synthesis in VACV-infected cells. HeLa H2B-mCh cells were mock infected or infected with VACV at MOI 8 and pulse labelled with the nucleotide analogue EdC (10 μM) for 15 min prior to fixation. Incorporated EdC was detected using the Click-iT^TM^ AF488 Imaging Kit and imaged by confocal microscopy. Representative micrographs are shown. Scale bar = 20μm. Per condition and biological replicate, 150–450 nuclei were scored either as EdC positive, or negative and normalised to the respective mock-infected control. Data represent biological triplicates and are displayed as the mean ± S.E.M. (**B**) BSC40 cells were infected with WT HA-D5 VACV. (**C**) The requirement for viral entry was tested by infecting BSC40 cells with recombinant VACV L1(+) and L1(−) VACV. (**D**) To test if viral entry was required, BSC40 cells were infected VACV C*ts*24 and incubated at either the permissive (31 °C), or non-permissive temperature (40 °C).

**Figure 4 viruses-14-00431-f004:**
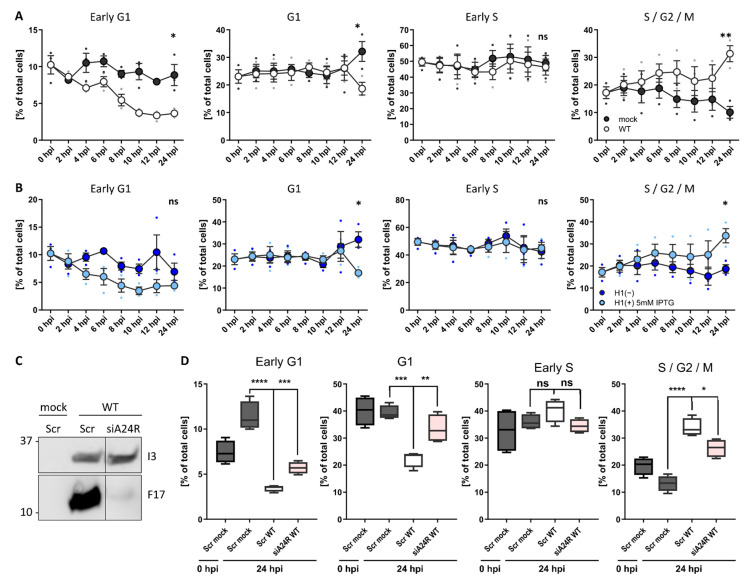
VACV post-replicative gene expression shifts the host cell cycle. (**A**) Cell cycle distribution of WT VACV-infected HeLa cells compared to uninfected controls. HeLa FUCCI cells were either mock infected or infected with WT VACV at MOI 2 and samples were harvested between 0 and 24 h. Cells were analysed by flow cytometry and the percentage G1, G1, early S, S/G2/M were assessed. Data represent biological triplicates and are displayed as the mean ± S.E.M. (**B**) Cell cycle distribution of VACV H1(−)-infected HeLa cells compared to parental H1(+)-infected controls. Samples were treated and analysed as in (**A**). Experiments represented in panel (A) and (B) were performed in parallel and share the same mock control. (**C**,**D**) To test if viral post-replicative gene expression was required to shift the cell cycle, the viral transcription factor A24 was silenced by siRNA. HeLa FUCCI cells were reverse transfected either with a scrambled control siRNA (Scr), or with siRNA targeting the viral transcription factor A24 (siA24R, rose). Samples were infected with WT VACV at MOI 1 and harvested at 24 hpi. (**C**) Inhibition of viral post-replicative gene expression was validated by immunoblot analysis of the viral early gene I3, and the viral late gene F17. (**D**) The cell cycle distribution was determined by flow cytometry and compared to Scr mock- (grey) and WT VACV (white)-infected control samples. Percentage of HeLa FUCCI cells in early G1, G1, early S, and S/G2/M. Data represent four biological replicates and are displayed as box (min to max), mean (line), and whiskers, indicating the 1 to 99 percentiles. Parametric, unpaired, two-tailed t-test for significance. ns. *p* > 0.05, * *p* < 0.033, ** *p* < 0.0021, *** *p* < 0.0002, and **** *p* < 0.0001.

**Figure 5 viruses-14-00431-f005:**
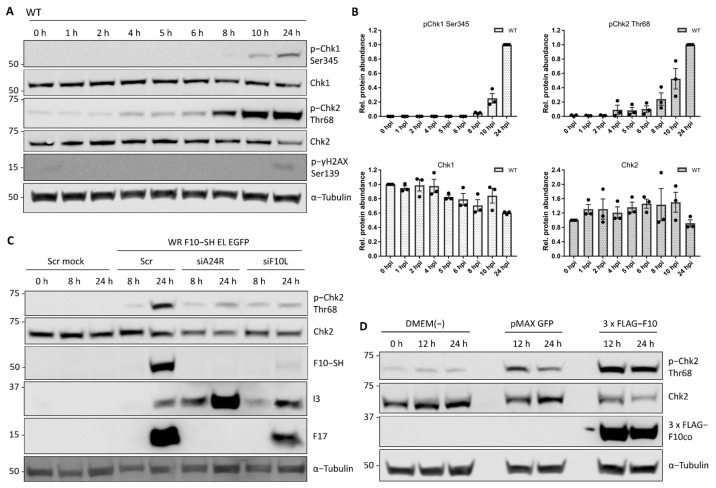
The viral kinase F10 activates the cellular DNA damage response. DDR activation was measured by phosphorylation of the DDR effector kinases Chk1 and Chk2, respectively. (**A**) HeLa cells were infected with WT VACV at MOI 5 and samples were harvested between 0 and 24 h. Whole-cell lysates were resolved via SDS-PAGE and immunoblotted for activating phosphorylation of Chk1 (Ser345), Chk1, activating phosphorylation of Chk2 (Thr 68), phosphorylated γH2AX (Ser139) which marks cellular DNA damage, and α-Tubulin as loading control. A representative blot of biological triplicates is shown. (**B**) Total and phosphorylated protein abundance was quantified and normalised to the loading control. Data represent biological triplicates, normalised to 0 h, and 24 h, respectively, and are displayed as the mean ± S.E.M. Black dots represent individual biological replicates. (**C**) The late viral kinase F10 is required to activate the cellular DDR. HeLa cells were reverse transfected with scrambled control siRNA (Scr), siA24R, or siF10L. At 48 h after transfection, cells were either mock infected, or infected with WR F10-SH EL EGFP (MOI 1) and samples were harvested at 0, 8, and 24 hpi. Whole-cell lysates were resolved via SDS-PAGE and immunoblotted for pChk2 (Thr68), Chk2, HA to detect expression of viral F10-SH, the early viral protein I3, the late viral protein F17, and α-Tubulin as loading control. A representative blot is shown. (**D**) The late viral kinase F10 is sufficient to activate the cellular DDR. HeLa cells were transfected with either a DMEM(-) control, pMAX GFP control vector, or codon-optimised 3 × FLAG-F10co. Samples were harvested at 0 h, 12 h, and 24 h post-transfection. Whole-cell lysates were resolved via SDS-PAGE and immunoblotted for activating phosphorylation of Chk2 (Chk2 Thr68), Chk2, FLAG to monitor expression of 3 × FLAG-F10co, and α-Tubulin as loading control. A representative blot of three biological replicates is shown.

**Figure 6 viruses-14-00431-f006:**
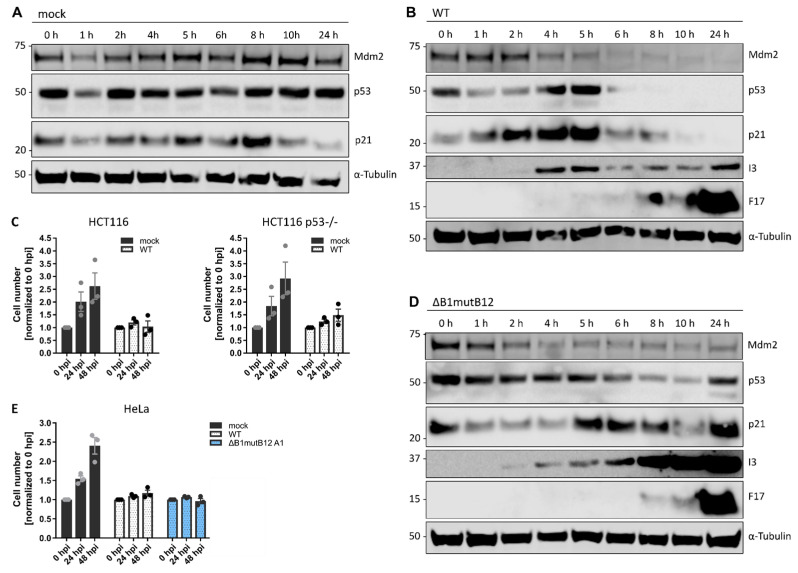
VACV infection degrades the DDR effectors Mdm2, p53 and p21. (**A**,**B**,**D**) HeLa cells were either mock infected, or infected with either WT VACV, or VACV ΔB1mutB12 at MOI 5 and harvested between 0 and 24 h. Whole-cell lysates were resolved via SDS-PAGE and blotted for Mdm2, p53, p21, the viral early prtein I3, the viral late protein F17, and α-Tubulin as loading control. Representative immunoblots of three biological replicates are shown. (**C**) VACV inhibits cell proliferation independent of p53. WT HCT116 and HCT116 p53-/- cells were either mock infected or infected with WT VACV at MOI 5 and cells were counted over 48 h. Cell numbers are normalised to 0 h and represent biological triplicates. Data are displayed as the mean ± S.E.M. (**E**) VACV inhibits cell proliferation independent of B1 and B12. HeLa cells were either mock infected, or infected with either WT VACV, or VACV ΔB1mutB12 at MOI 5. Samples were processed and analysed as in (**C**). HeLa mock and WT data are reproduced from Figure 1A.

**Figure 7 viruses-14-00431-f007:**
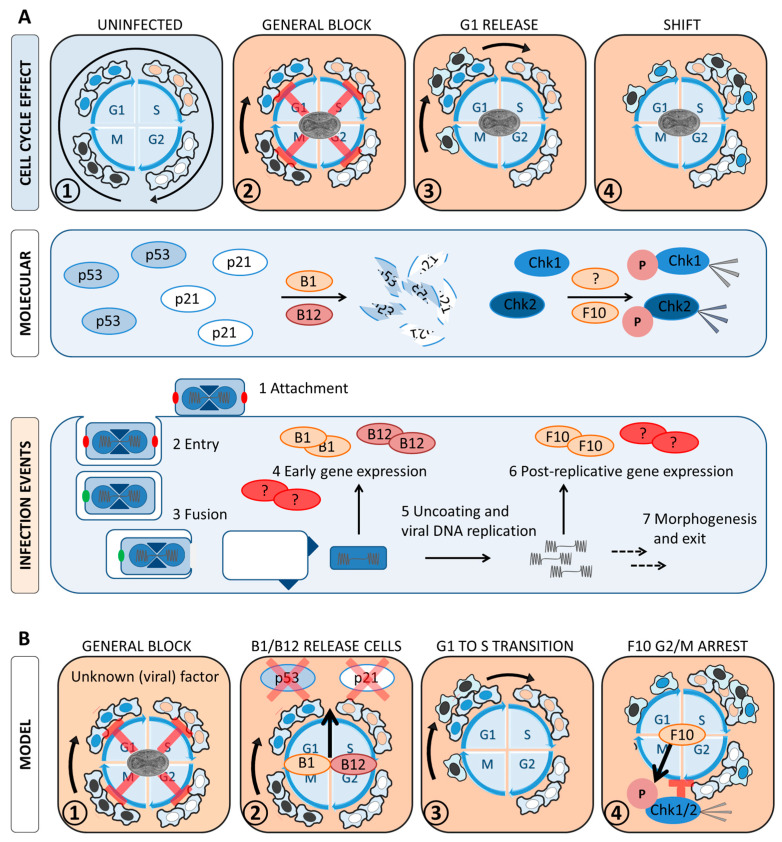
Model of VACV-induced cell cycle changes. (**A**) Experimental observations. VACV infection establishes a general cell cycle arrest early during infection. Progressing infection then selectively releases cells from the G1 block, which causes a cell cycle shift into S/G2. While the mechanism and cellular factors that regulate these changes are unknown, our data indicate that a viral early gene arrests the cell cycle and we show that the cell cycle shift is mediated by a post-replicative viral gene. We further demonstrate that the viral kinase F10 activates the cellular DNA damage response (DDR), while viral B1/B12 cause degradation of the DDR effector p53 and its transcriptional target p21. This observation is in line with the model that arrested, infected cells are released from the G1 block by inactivation of p53 and p21. (**B**) Model of VACV-induced cell cycle changes. An early infection event inhibits the cell cycle (i.e., cells are frozen in their current cell cycle stage). After this initial block, B1 and/or B12 facilitate degradation of p53 and p21, removing the G1/S checkpoint, allowing a shift in cell cycle distribution from G1 through S to G2. This event coincides with F10-directed phosphorylation of Chk1 and 2, triggering the G2/M checkpoint which serves to keep the infected cells at the border of G2 and M. Rounded black arrows represent progression through the cell cycle; straight black arrows indicate the next stage in the virus life cycle: red arrows indicate inhibition of the respective cell cycle stage; red ellipsoids indicate an unknown cellular and/or viral protein; starburst from Chk1 and Chk2 indicate kinase activation.

## Data Availability

The data presented in this study are available on request from the corresponding author.

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
