# Peer review of "Vaccinia Virus Arrests and Shifts the Cell Cycle"

_viruses, 2022, doi:10.3390/v14020431_

Round 1
Reviewer 1 Report
In the study of Martin et al. entitled “Vaccinia Virus Arrests and Shifts the Cell Cycle”, the authors investigate the effect of vaccinia virus (VACV) infection on cell proliferation and host cell cycle progression. For this, was used a subset of VACV mutants after characterizing the stage of infection required for inhibition of cell proliferation and defining the viral effectors required to dysregulate the host cell cycle. Consistent with previous studies, the results show that VACV inhibits, and subsequently shifts the host cell cycle. Was demonstrated that these two phenomena are independent of one another, with viral early genes being responsible for cell cycle inhibition, and post-replicative viral gene(s) responsible for the cell cycle shift. Extending previous findings, the work shows that the viral kinase F10 is required to activate the DNA damage checkpoint and that the viral B1 kinase and/or B12 pseudokinase mediate degradation of checkpoint effectors p53 and p21 during infection. Thus, the authors conclude that VACV modulates host cell proliferation and host cell cycle progression through a temporal expression of multiple VACV effector proteins. The study is important for analyzing factors of the virus-host interface and should be accepted after minor revisions.
Minor revisions:
1) In general, the article is well written, although the abstract does not reflect all the results obtained in the study, because it is difficult to summarize all the stages of the study in a restricted number of words. Furthermore, the results are supported by different and subsequent experiments that together prove the hypotheses raised during the execution of the work.
2) Page 1, line 60: “To date, the only VACV protein connected to cell cycle 60 regulation is the viral kinase B1 [27].” So, does this mean that determining the effect of viral F10 kinase and viral B12 pseudo kinase is the major contribution of this study?
3) Page 2, line 81: define the acronyms: topoisomerase II binding protein 1 (TopBP1) and proliferating cell nuclear antigen (PCNA). Line 82: serine/threonine-protein kinase also known as ataxia telangiectasia and Rad3-related protein (ATR) and checkpoint kinase 1 (Chk1). Line 88: minichromosome maintenance complex (MCM).
4) Page 3, lines 129 to 131: Add the function of inhibitors, eg Mevinolin: a highly potent competitive inhibitor of hydroxymethylglutaryl-coenzyme A reductase (10.1073/pnas.77.7.3957). This can be added as supplemental material.
5) Page 3, lines 136 to 140: The same for the description of each recombinant.
6) Page 3, page 144: What do MV stocks means?
7) Page 5, line 230: The cell proliferation definition should be added in the introduction, as well as the differentiation for host cell cycle progression. For the reader seems to be the same thing.
8) Page 7, line 294: Is there no significant difference between mock- and VACV-infected-cells in times 5, 6, 10, and 24 h in Figure 2B?
9) Page 11: Figure 4A and 4B: change color or use circles with and without colors to facilitate the differentiation between mock- and WT-infected cells.
10) The greatest difficulty lies in determining which viral genes act in each of the stages studied. For example, which acts specifically on cell proliferation and which acts on host cell cycle progression? Both stages and the viral infection occur at the same time and it is difficult to separate this even didactically. For example, viral kinase F10, viral B1 kinase, and viral B12 pseudokinase act on which of these steps?
11) In Figure 7 or in a new figure, can the authors add the cell cycle steps pointing out which genes (early or late) and which viral kinases (F10, B1, and B12) act in exactly which phases? This will make it easier for the reader to understand.
Reviewer 2 Report
In this study Martin et al., elucidated the mechanism corresponding to vaccinia virus (VACV) and its influence in altering host cell cycle and activate DNA damage response. Further, they demonstrated DNA damage checkpoint is activated by F10 Kinase and other viral kinases B1 and B12) degrade host p53 and p21. This article is discussed the undermined aspects of this disease. However, authors need to address the following concerns before publication.
- “hpi" abbreviation should be included in the method section 2.4. Also, MOI abbreviation can be added, at the relevant text.
- In Figure 1B, what was the rationale behind using MOI 5 and MOI 10. Were authors expecting a dose dependent % LDH activity from this study?
- Mock control in Figure 4B were missing?
- In figure 6, description of study related to figure 6 D is missing in the figure legend.
- The manuscript can be revised further for grammatical and typological errors.
Author Response
We thank the reviewer for their constructive comments and suggestions. In line with these, we have now revised the text and performed a careful reading of the final manuscript for editorial errors as outlined in the point-by-point below.
Reviewer 2:
In this study Martin et al., elucidated the mechanism corresponding to vaccinia virus (VACV) and its influence in altering host cell cycle and activate DNA damage response. Further, they demonstrated DNA damage checkpoint is activated by F10 Kinase and other viral kinases B1 and B12) degrade host p53 and p21. This article is discussed the undermined aspects of this disease. However, authors need to address the following concerns before publication.
1-“hpi" abbreviation should be included in the method section 2.4. Also, MOI abbreviation can be added, at the relevant text.
Thank you for pointing this out. We have defined hpi (hours post infection) and MOI (multiplicity of infection in the text.
2-In Figure 1B, what was the rationale behind using MOI 5 and MOI 10. Were authors expecting a dose dependent % LDH activity from this study?
We used a MOI 5 for the actual proliferation assays throughout the manuscript. We utilized MOI 5 AND MOI 10 for the LDH assay to highlight that even at high levels of infection cell death was not the driving factor for the reduction in cell proliferation. We have modified the text to this effect.
3-Mock control in Figure 4B were missing?
Thank you for pointing this out. Experiments shown in 4A and 4B were done together i.e. the mock in panel 4A is also the mock for 4B. We have added this information to the figure legend.
4-In figure 6, description of study related to figure 6 D is missing in the figure legend.
We have added the description for panel D to the figure legend.
5-The manuscript can be revised further for grammatical and typological errors.
We have performed a careful reading of the manuscript to correct the errors.
Reviewer 3 Report
Martin et al., extensively worked on vaccinia virus'(VACV) cell cycle regulation. They used wild type and mutated strains of VACV to study the impact of viral genes on host cell cycle.
For methodology, the authors used reasonable cell lines and techniques to test their ideas.
The results confirmed viral kinases F10, B1 and B12 pseudo kinase affect the host cell cycle independently. F10 is responsible for DNA damage checkpoint while B1 kinase and/or B12 pseudokinase involve in degradation of p53 and p21.
This work can be considered for the publication.
Author Response
We thank the reviewer for their constructive comments and suggestions. In line with these, we have now revised the text and performed a careful reading of the final manuscript for editorial errors as outlined in the point-by-point below.
Reviewer 3:
Martin et al., extensively worked on vaccinia virus'(VACV) cell cycle regulation. They used wild type and mutated strains of VACV to study the impact of viral genes on host cell cycle.
For methodology, the authors used reasonable cell lines and techniques to test their ideas.
The results confirmed viral kinases F10, B1 and B12 pseudo kinase affect the host cell cycle independently. F10 is responsible for DNA damage checkpoint while B1 kinase and/or B12 pseudokinase involve in degradation of p53 and p21.
This work can be considered for the publication.
We thank the reviewer for their comments.